# Neurophysiological Mechanisms of Resilience as a Protective Factor in Patients with Internet Gaming Disorder: A Resting-State EEG Coherence Study

**DOI:** 10.3390/jcm8010049

**Published:** 2019-01-06

**Authors:** Ji-Yoon Lee, Jung-Seok Choi, Jun Soo Kwon

**Affiliations:** 1Department of Psychiatry, SMG-SNU Boramae Medical Center, Seoul 07061, Korea; idiyuni91@gmail.com; 2Interdisciplinary Program in Neuroscience, Seoul National University College of Natural Sciences, Seoul 08826, Korea; kwonjs@snu.ac.kr; 3Department of Psychiatry, Seoul National University College of Medicine, Seoul 03080, Korea

**Keywords:** Internet gaming disorder, resilience, resting-state electroencephalography (EEG), coherence, moderated mediation

## Abstract

Background: Resilience, an important protective factor against Internet gaming disorder (IGD), is the ability to recover from negative emotional experiences and constitutes a flexible adaptation to stress. Despite the importance of resilience in predicting IGD, little is known about the relationships between resilience and the neurophysiological features of IGD patients. Methods: We investigated these relationships using resting-state electroencephalography (EEG) coherence, by comparing IGD patients (*n* = 35) to healthy controls (*n* = 36). To identify the resilience-related EEG features, the IGD patients were divided into two groups based on the 50th percentile score on the Connor–Davidson Resilience Scale: IGD with low resilience (*n* = 16) and IGD with high resilience (*n* = 19). We analyzed differences in EEG coherence among groups for each fast frequency band. The conditional indirect effects of resilience were examined on the relationships between IGD and resilience-related EEG features through clinical symptoms. Results: IGD patients with low resilience had higher alpha coherence in the right hemisphere. Particularly, resilience moderated the indirect effects of IGD on alpha coherence in the right hemisphere through depressive symptoms and stress level. Conclusion: These neurophysiological findings regarding the mechanisms underlying resilience may help to establish effective preventive measures against IGD.

## 1. Introduction

The Internet has quickly become essential to our daily lives because it constitutes an endless source of information and entertainment. In particular, the playing of Internet games has rapidly expanded commensurate with the increased use of the Internet. Because these games provide unlimited fun and entertainment, many Internet users, including children and adolescents, are tempted to play them. However, excessive use of the Internet can be conceptualized as an addictive behavior [1].

Persistent and recurrent use of Internet games resulting in psychological impairments or distress is defined as Internet gaming disorder (IGD) by the Diagnostic and Statistical Manual of Mental Disorders, Fifth Edition (DSM-5) [2]. Currently, IGD is a widespread problem that affects individuals worldwide [3]. Due to the rapid increase in the number of IGD diagnoses, a comprehensive understanding of the disorder is necessary to develop effective strategies for its prevention. Furthermore, clarification of the neural mechanisms underlying the risk and protective factors for IGD will aid in the development of such strategies.

Several protective factors for, and risk factors for the development and maintenance of, IGD have been identified. A meta-analysis of studies on Internet addiction (IA) in Korea revealed that various factors increase the likelihood of the disorder [4]. For example, IA has positive associations with control- and regulation-related factors (e.g., attention problems), temperament factors (e.g., addiction and absorption traits), emotion- and mood-related factors (e.g., anger and aggression, and depression and anxiety), and coping factors (e.g., negative stress coping). Additionally, psychiatric disorders such as attention deficit hyperactivity disorder (ADHD), depression, social phobia, and hostility predict the emergence of IA [5]. In particular, patients with alexithymia reported the high likelihood of IA. They had difficulty in identifying feelings, higher dissociative experiences, lower self-esteem, and higher impulse dysregulation were associated with higher emergence of IA [6].

Recently, researchers and clinicians have shifted their focus from risk factors for IGD to protective factors for the disorder, with the aim of preventing its occurrence and aiding the recovery of IGD patients. A review of longitudinal studies on adolescents and young adults demonstrated that higher levels of self-control and self-esteem, in addition to satisfaction of basic psychological needs, protect against problematic Internet use [7]. Choi et al. [8] reported that character strength of courage indicating encompasses vitality and bravery has a negative relationship with IA.

Meanwhile, resilience is receiving attention because it regarded as one of the prominent protective factors for IGD. Resilience can be defined as the personal qualities that enable one to thrive in the face of adversity throughout life [9]. Resilience involves rapid recovery from mental health disturbances following exposure to adversity [10]. Resilience plays an important role in life satisfaction [11]. Resilience is an important protective factor for the pathophysiology of addictive disorders [12] and is a strong protective factor against IGD. A previous study showed that higher the level of the participants’ resilience assessed by Connor–Davidson Resilience Scale (CD-RISC), the lower the level of their IA [13,14]. In a multiple mediation study with children showed that resilience was strongly associated with IA directly, and indirectly, through peer relationship and depression. This study suggests that resilience is as a predictor of IA [15].

At present, there is a lack of research investigating the relationship between resilience and IGD and, importantly, the precise neural correlates that underlie resilience remain largely unknown. Empirical studies of resilience have focused exclusively on the behavioral and psychosocial correlates of, and contributors to, the phenomenon however, the biological correlates and/or contributors of resilience have yet to be fully explored [16]. However, Curtis and Cicchetti [17] noted the importance of examining biological factors that contribute to resilience and suggested that adopting a biological perspective is one facet of an all-encompassing approach to understanding resilience. Thus, novel insights into the neural mechanisms of resilience will increase understanding of this concept.

Over the past few decades, neuroimaging methods have become an increasingly important tool to study the neural correlates of adaptive and non-adaptive behaviors [18]. Although there has been a recent increase in neuroimaging studies assessing the neural mechanisms that underlie resilience or IGD, respectively, little is known about the neural correlates of resilience in IGD patients. Thus, the present study used electroencephalography (EEG) to explore the neurophysiological correlates underlying the relationship between resilience and IGD. EEG has several advantages relative to other neuroimaging tools, including a high temporal resolution, non-invasiveness, and a significantly lower cost. EEG involves electrophysiological recordings that show electrical activity in the brain, thus allowing baseline measurement of the brain state prior to information processing [19]. Resting-state EEG activities were recorded because these could enhance the current understanding of basic brain function in IGD and it has been proposed that resting-state EEG is not merely an epiphenomenon, but rather reflects the electrophysiological underpinnings of human behavior [20].

EEG coherence is primarily a measure of phasic synchrony [21], and is widely considered to indicate the functional interaction or connectivity between two brain regions. Coherence has proved useful for detecting brain pathologies in a variety of conditions associated with neuronal dysfunction [22]. Several studies have investigated the neural mechanisms of IGD using EEG coherence. Our previous study showed that patients with IGD and alcohol use disorder exhibit different neurophysiological patterns of brain connectivity, and that increases in the fast-phasic synchrony of gamma coherence might be a core neurophysiological feature of IGD [23]. In a 6-months follow-up study, IGD patients exhibited increased beta and gamma intrahemispheric coherence and increased delta intrahemispheric coherence of the right hemisphere than healthy controls (HC) at baseline. However, these abnormal phase correlation patterns are not normalized after 6 months of outpatient management, even though the patients with IGD patients showed significant improvements in their IGD symptoms [24]. Furthermore, a study compared major depressive disorder (MDD) patients with IGD and without IGD showed that MDD with IGD group have increased beta intrahemispheric coherence values within the right frontal-temporal, temporal-occipital, and parietal-occipital areas compared with the only MDD patients [25].

The neural correlates of resilience are also associated with specific psychological and/or cognitive phenomena, as well as psychiatric disorders including emotion dysregulation, post-traumatic stress disorder (PTSD). A study of PTSD patients indicated that higher right prefrontal theta power during the first and last rapid eye movement (REM) periods in resilient normal participants compared with participants with PTSD [26]. Additionally, it was reported that EEG asymmetry across central cortical regions distinguishes between high- and low-resilience children; specifically, left hemisphere activity is characteristic of maltreated children who exhibit resilience [27].

To our knowledge, no studies have investigated the neurophysiological features of individuals with IGD and their relationship with resilience. Thus, the present study aimed to determine the neurophysiological correlates of resilience, via resting-state EEG coherence analysis, in IGD patients and HC. Based on previous studies, in the present study it was hypothesized that the resting-state EEG phasic synchrony of fast waves in IGD patients and HC would be dissociable according to the degree of resilience; the distinct EEG coherence features of IGD and HC would be mediated by resilience-related clinical symptoms, in which resilience would moderate the indirect effect of IGD on the distinct EEG features through clinical symptoms.

Particular emphasis was placed on the three fast wave bands (alpha, beta, and gamma) that were shown to be related to IGD in our previous studies [23,28,29]. We selected depressive symptoms and stress level as hypothesized mediators that had significant neurobiological relationships with resilience [10]. Furthermore, the previous study showed that resilience had moderating effect on the association between IGD and negative psychological affects [15,30]. Hypothetical model of the present study was presented in Figure 1.

## 2. Materials and Methods

### 2.1. Participants

The present study initially included 76 male adults, aged between 18 and 40 years, who were recruited from SMG-SNU Boramae Medical Center in Seoul, South Korea, and the surrounding community. None of the participants had a history of significant head injury, seizure, intellectual disability [Intelligence quotient (IQ) ≥ 80], or psychotic or neurological disorders. Additionally, all participants were medication-naive and right-handed. IGD was diagnosed by trained clinicians based on DSM-5 criteria (Appendix A); participants who spent more than 4 hours per day and 30 hours per week playing Internet games were included in the IGD group. Young’s Internet Addiction Test (IAT) was used to assess the severity of IGD. All HC were recruited from the local community and universities, none had a history of any psychiatric disorder, and all played Internet games for less than 2 hours per day.

The data of five participants were not included in the final analyses, either because their EEG recordings were disrupted by excessive movement or the EEG channels were of poor quality (>100 µV^2^ Sq). Thus, a total of 71 participants were included in the present analyses and categorized as IGD (*n* = 36) or HC (*n* = 35).

### 2.2. Clinical Assessments

#### 2.2.1. CD-RISC

Resilience was assessed using the CD-RISC [9], which is a 25-item self-report instrument that uses 5-point response scales, as follows: 0 = not true at all, 1 = rarely true, 2 = sometimes true, 3 = often true, and 4 = true nearly all of the time. The CD-RISC captures how the participant felt over the past month and total scores range from 0–100, with higher scores reflecting greater resilience. The Cronbach alpha coefficient was 0.967.

#### 2.2.2. Young’s IAT

The severity of IGD was evaluated with Young’s IAT [31,32], which includes 20 items rated on 5-point scales, with the total possible score thus ranging from 20–100.

#### 2.2.3. Beck Depression Inventory-II (BDI)

Beck depression inventory-II (BDI) has 21 items measuring the severity of depressive symptoms during the past 2 weeks [33]. Items are four points Likert scale from 0 to 3 and total scores range from 0 to 63 with higher scores indicating higher depressive symptoms.

#### 2.2.4. Psychosocial Well-Being Index (PWI)

Stress level was measured with a Psychosocial Well-Being Index (PWI) which contains 45 items [34]. PWI was developed on the basis of the General Health Questionnaire generated by Goldberg [35], which was designed to evaluate psychological stability among community populations and was subsequently modified to meet the characteristics of Korean populations [34]. It contains questions about physical and psychological status over the last few weeks, covering social role performance, self-confidence, depression, sleep disturbance, anxiety, and the general well-being of respondents. Score range from 0 to 135 with higher scores indicating higher distress symptoms.

#### 2.2.5. Wechsler Adult Intelligence Scale, Korean Version

The Korean version of the Wechsler Adult Intelligence Scale [36] was administered to all participants to estimate IQ levels.

### 2.3. EEG Recoding

Each participant was seated on a comfortable chair in an isolated sound-shielded room with dimmed lights, and then underwent EEG recording in a resting state that lasted for 10 min (4 min with eyes closed, 2 min with eyes open, and 4 min with eyes closed). All participants were instructed to avoid moving or becoming drowsy. All EEG activity was recorded using a 64-channel Quik-cap (Compumedics Neuroscan, El Paso, TX, USA) based on the modified international 10/20 system, in conjunction with vertical and horizontal electrooculograms (EOGs) and one bipolar reference electrode connected to the mastoid. All EEG acquisitions were done using SynAmps 2 (Compumedics, Abbotsford, Australia) and the Neuroscan system (Scan 4.5; Compumedics). EEG signals were amplified at a sampling rate of 1000 Hz using a 0.1–100 Hz online bandpass filter and a 0.1–50 Hz offline bandpass filter, while electrode impedance was kept below 5 kΩ.

All acquired EEG data were processed with NeuroGuide software (ver. 2.6.1; Applied Neuroscience, St. Petersburg, FL, USA). For the analyses, 19 of the 64 channels were selected according to the montage set with linked ear references from the NeuroGuide, as follows: FP1, F3, F7, Fz, FP2, F4, F8, T3, C3, Cz, T4, C4, T5, P3, O1, Pz, T6, P4 and O2. All EEG recordings under eyes-closed conditions were selected and artifacts were removed using the artifact rejection toolbox in NeuroGuide and based on visual inspection. Finally, the artifact-free epochs, ranging in length from 24.09–323.99 s, were averaged for each electrode. The average epoch length was 124.59 ± 66.92 s in the IGD group and 139.72 ± 52.30 s in the HC group; one-way analysis of variance (ANOVA; *p* = 0.291) revealed that the epoch lengths of the two groups did not differ significantly. Visualizations were performed with MATLAB software (MathWorks, Natick, MA, USA).

### 2.4. Coherence Analyses

Coherence values were computed for all pairwise combinations of the 19 channels, for each of the three frequency bands (alpha, beta, and gamma), using NeuroGuide software [37]. The computation of coherence was based on previous studies [23,38]. A total of 171 intrahemispheric and interhemispheric pairwise combinations of electrodes were obtained and intrahemispheric coherence was examined using the F3-C3, F3-T3, F3-P3, C3-T3, C3-P3, and T3-P3 electrode pairs in the left hemisphere and the F4-C4, F4-T4, F4-P4, C4-T4, C4-P4, and T4-P4 electrode pairs in the right hemisphere. Interhemispheric coherence was calculated between electrode pairs F3-F4, C3-C4, T3-T4, and P3-P4 [39].

### 2.5. Statistical Analyses

To identify the effects of resilience on EEG activity in IGD patients, the IGD group was divided into two subgroups based on the degree of resilience, as determined by the CD-RISC: IGD patients with low resilience (LowRe-IGD, *n* = 16; score range: 8–43, which was below the 50th percentile) and IGD patients with high resilience (HighRe-IGD, *n* = 19; score range: 44–49, which was above the 50th percentile). Subsequently, we conducted to determine the EEG correlates of resilience.

Comparisons of demographic and clinical variables among the groups (IGD vs. HC and LowRe-IGD vs. HighRe-IGD vs. HC) were performed using one-way ANOVA. Next, a generalized estimating equation (GEE) has been conducted to analyze the characteristics of EEGs in each band. GEEs are employed to consider unknown and possible correlations between repeated or multiple outcomes from the same subjects [40,41]. In the absolute power analyses, group (LowRe-IGD, HighRe-IGD, HC), region (frontal, central, posterior), hemisphere (left, right), and their interaction effects were tested in each band with a GEE. In the coherence analyses, intrahemispheric and interhemispheric coherence values were assessed according to the following factors: (1) intrahemispheric coherence: group (LowRe-IGD, HighRe-IGD, HC) × region (fronto-central, fronto-temporal, fronto-parietal, centro-temporal, centro-parietal, temporo-parietal) × hemisphere (left, right) and (2) interhemispheric coherence: group (LowRe-IGD, HighRe-IGD, HC) × region (frontal, central, temporal, parietal). Years of education and IQ were adjusted for because these characteristics differed among the groups. Furthermore, we performed Pearson’s correlation analyses to determine the relationships among resilience (CD-RISC scores), clinical data (depressive symptoms and stress level), and EEG features that showed significant main or interaction effects in GEE analyses. All statistical analyses were performed using SPSS software (version 23.0, IBM, Armonk, NY, USA).

The PROCESS macro for SPSS [42], a bootstrapping approach, was employed to analyze the mediating effects, moderating effects, and conditional indirect effects (moderated mediation). First, simple mediation models (model 4) were tested to determine whether clinical features mediated the relationship between IGD on the EEG feature which was correlated with resilience. Second, simple moderation models (model 1) were employed to confirm how IGD interacted with resilience and clinical symptoms. Third, moderated mediation models (model 7) were evaluated to determine whether the indirect effect of IGD on the distinct EEG features through the depressive symptoms and stress level were moderated by resilience. Significance was determined using bias-corrected 95% confidence intervals (BCa CI) by bootstrapping using 5000 resamples, and it was indicated when CI did not include zero. We used the Sobel test (Z) to examine the indirect effect size of mediating models. Based on the pick-a-point method, conditional effects at plus and minus one standard deviation around the mean of resilience were estimated [42]. Continuous variables were mean-centered to reduce any multicollinearity, and IQ and years of education were controlled in mediation, moderation, and moderated mediation analyses.

For all analyses, *p*-values < 0.05 were considered to indicate statistical significance. When Bonferroni-corrected post-hoc comparisons were performed, *p*-values < 0.0167 were considered to indicate statistical significance.

### 2.6. Ethics

The institutional review board of SMG-SNU Boramae Medical Center approved the study protocol, which adhered to the principles of the Declaration of Helsinki. All participants understood the study procedure and provided written informed consent prior to participation.

## 3. Results

### 3.1. Demographic and Clinical Data

The demographic and clinical characteristics among LowRe-IGD, HighRe-IGD and HC groups are shown in Table 1. CD-RISC score significantly differed among the groups, where the LowRe-IGD group had a lower score than the HighRe-IGD and HC groups (LowRe-IGD: 26.00 ± 12.81; HighRe-IGD: 63.26 ± 9.02; HC: 74.14 ± 8.44; *p* < 0.001). The severity of IGD in the LowRe-IGD group was significantly higher than that in the HighRe-IGD and HC groups (*p* < 0.001). In terms of clinical data, LowRe-IGD showed lower BDI and PWI scores than HighRe-IGD and HC (all *p* < 0.001). The demographic and clinical data of the IGD and HC groups are compared in Appendix A.

### 3.2. Intrahemispheric Coherence

Analyses of intrahemispheric coherence revealed that there were significant group (*x*^2^ = 13.515, *p* = 0.001) and group × hemisphere (*x*^2^ = 7.358, *p* = 0.025) interaction effects for the alpha band (Table 2 and Figure 2). The LowRe-IGD group showed greater alpha intrahemispheric coherence than the HighRe-IGD (*p* = 0.001) and HC (*p* = 0.009) groups. Additionally, the higher alpha coherence in the LowRe-IGD group was primarily evident in the right hemisphere (HighRe-IGD: *p* < 0.001; HC: *p* = 0.001). 

Although there was a significant group × region interaction effect for beta intrahemispheric coherence (*x*^2^ = 26.246, *p* = 0.001), there were no group differences (all *p* > 0.0167). Regarding intrahemispheric coherence in the gamma band, no significant main or interaction effects were found.

### 3.3. Interhemispheric Coherence

The interhemispheric coherence data was shown at the bottom of Table 2 and Figure 2. In the alpha band, there was a significant main effect of group (*x*^2^ = 9.836, *p* = 0.007), where the LowRe-IGD group exhibited higher alpha coherence compared to the HC group (*p* = 0.008).

There was no group × region interaction effect in the alpha band. In the beta and gamma bands, there were no main effects of group, and no group × region, group × hemisphere, or group × region × hemisphere interaction effects for interhemispheric coherence.

### 3.4. Correlation Analyses

Correlation analyses of CD-RISC scores and coherence that had significant group differences in the GEE analyses was conducted (alpha intrahemispheric coherence in all electrode pairs and in the right hemisphere, and alpha interhemispheric coherence; Appendix A). The mean value of each EEG variable was included in the correlation analysis.

In the overall cohort, there was a significant relationship between CD-RISC score and alpha coherence in the right hemisphere (*r* = −0.264, *p* = 0.026); however, this relationship differed between the IGD patients and HC, where the IGD group showed a negative correlation (*r* = −0.435, *p* = 0.009) and the HC group a positive correlation (*r* = 0.440, *p* = 0.007). There were significant positive correlations between BDI and alpha coherence in IGD patients (all electrode pairs: *r* = 0.373, *p* = 0.027; right hemisphere: *r* = 0.417, *p* = 0.013), whereas HC group had no significant correlations (all electrode pairs: *r* = 0.026, *p* = 0.880; right hemisphere: *r* = 0.114, *p* = 0.509). PWI had positive correlations with alpha coherence in the right hemisphere in the overall cohort (*r* = 0.318, *p* = 0.007) and IGD patients (*r* = 0.461, *p* = 0.005).

### 3.5. Tests of Mediation

We examined the mediating roles of depressive symptoms and stress level on the relationship between IGD and alpha coherence in the right hemisphere via PROCESS macro. The simple mediation analysis with depressive symptoms as a mediator showed that IGD had no total effect (B = 3.885, S.E = 3.452, *p* = 0.264) and direct effect (B = −2.806, S.E = 3.664, *p* = 0.446) on alpha coherence in the right hemisphere. However, the effect of IGD to depressive symptoms (B = 15.571, S.E = 2.094, *p* < 0.001) and that of depressive symptoms to alpha coherence in the right hemisphere had positive associations (B = 0.580, S.E = 0.159, *p* = 0.001). The Sobel test indicated significant indirect effect of depressive symptoms (Z = 3.254 *p* = 0.001). It was confirmed by the bootstrap results with 5000 resamples, with indirect effect of 9.031 and 95% bias-corrected and accelerated bootstrap interval (BCa CI) around this value not containing zero [4.042, 16.109].

In the model with stress level as a mediator, the total effect (B = 3.885, S.E = 3.452, *p* = 0.264) and direct effect were not significant (B = −2.227, S.E = 3.840, *p* = 0.564). However, a significant effect of stress level regressed on IGD (B = 39.425, S.E = 5.496, *p* < 0.001) and of alpha coherence in the right hemisphere regressed on stress level (B = 0.179, S.E = 3.840, *p* = 0.004) were found. The Sobel test indicated significant indirect effects of stress level (Z = 2.757, *p* = 0.006). Bootstrap results confirmed this indirect effect of 7.064 and BCa CI around [1.994, 14.007]. In other words, the simple mediation analysis yielded the evidence of a positive indirect effect of IGD on alpha coherence in the right hemisphere through depressive symptoms and stress level, with IGD linked to more depressive symptoms and higher stress level related to greater alpha coherence in the right hemisphere.

### 3.6. Tests of Moderated Mediation

Before conducting moderated mediation analyses, the moderating effects of resilience on the relationship between IGD and depressive symptoms and stress level were evaluated. The model showed that interaction effects of resilience and IGD on clinical symptoms were significant (depressive symptoms: B = −0.444, S.E = 0.138, *p* = 0.002; stress level: B = −0.444, S.E = 0.138, *p* = 0.028). Significant conditional effects of IGD on clinical symptoms were observed at the low resilience (mean − 1 S.D; depressive symptoms: B = 19.959, boot S.E = 4.790, *p* < 0.001; stress level: B = 29.265, boot S.E = 10.363, *p* = 0.006) and moderate resilience (mean; depressive symptoms: B = 10.412, boot S.E = 2.440, *p* < 0.001; stress level: B = 14.890, boot S.E = 5.280, *p* = 0.006). On the other hand, there was no significant conditional effect of IGD on clinical symptoms in participants with high resilience (mean − 1 S.D; depressive symptoms: *p* > 0.05); i.e., severe depressive symptoms and higher stress level compared with HC were presented in the IGD with lower resilience and were not evident in the IGD with higher resilience (Figure 3).

Next, the moderated mediation analysis examined the potential conditional effects of resilience on the mediation process. A significant indirect effect was found for IGD on alpha coherence in the right hemisphere through depressive symptoms (Table 3). The overall moderated mediation model was supported with the index of moderated mediation (B = −0.258, S.E. = 0.094, BCa CI [−0.462, −0.093]). An effect of IGD on alpha coherence in the right hemisphere was not significant (B = −2.806, S.E = 3.664, *p* = 446). However, moderated mediation effect of depressive symptoms indicated that the mediating effect was conditional on alpha coherence in the right hemisphere. Depressive symptoms mediated IGD on alpha coherence in the right hemisphere at low (mean − 1 S.D; B = 11.576, boot S.E = 3.958, BCa CI [4.507, 19.983]) and moderate (mean; B = 6.039, boot S.E = 2.266, BCa CI [2.377, 11.375]) degree of resilience.

Moderated mediation model with stress level (mediator), it was significant that the indirect effect of IGD on alpha coherence in the right hemisphere through the stress level was moderated by resilience (Index of moderated mediation = −0.120, S.E. = 0.075, BCa CI [−0.326, −0.019]; Table 4). There were significant positive associations between IGD and alpha coherence in the right hemisphere through the stress level only at low (mean − 1 S.D; B = 5.244, boot S.E = 2.916, BCa CI [1.271, 13.060]) and moderate (mean + 1 S.D; B = 2.668, boot S. E. = 1.557, BCa CI [0.584, 6.901]) level of resilience.

## 4. Discussion

The primary goal of the present study was to investigate the relationship between resilience and IGD pathology via resting-state EEG functional connectivity analyses. IGD patients exhibited lower resilience than HC; furthermore, IGD patients with low resilience showed increased coherence, especially in the right hemisphere, compared to IGD patients with high resilience and HC. Importantly, there were conditional indirect effects of IGD on alpha coherence in the right hemisphere through depressive symptoms and level of stress, and those indirect effects were moderated by the degree of resilience.

The present study observed that indirect effects of IGD on alpha coherence in the right hemisphere through depressive symptoms and stress level were significant in IGD patients with lower and moderate resilience level. In other words, IGD patients with lower resilience showed severe depressive symptoms and higher stress level, which was related with greater alpha coherence in the right hemisphere. Resilience is typically studied in PTSD patients because this trait plays an important role in preventing the development of the disorder after marked acute or chronic stress [43]. An EEG study of PTSD patients found that individuals with PTSD have increased alpha coherence between the precuneus and right inferior parietal lobe, where these abnormal neurophysiological patterns reflect symptoms such as high persistent intrusive thoughts, difficulties in emotional regulation, fragmentation of traumatic memories, and working memory deficits. Furthermore, it is well known that IGD is associated with anxiety, depression, and stressful life events [44,45].

A study on children revealed that males who experienced abuse (psychological neglect, physical neglect, paternal physical violence, and/or sexual violence) in the past year were more likely than their peers to have relatively high levels of PTSD and IA [46]. Thus, increased alpha coherence in the right hemisphere of IGD patients may indicate that non-resilient individuals have trouble dealing with stress and negative emotions associated with unpleasant past events, which in turn would predispose them to addiction to Internet games. In other words, individuals may try to “hide” in the virtual world (Internet games) to avoid or reduce negative affect, or due to difficulties in coping with acute or chronic stress and negative emotions. Thus, clinicians should pay attention to dysfunctional emotional regulation and stress coping when treating IGD patients with lower resilience and their altered alpha coherence in the right hemisphere.

We found significant group effects on beta intrahemispheric coherence, whereas there were no group differences among three groups. It is not concordant with previous studies that reported IGD patients had reduced beta power and greater gamma power and coherence [23,24]. However, the inconsistent results with previous studies may be explained several reasons. First, the present study tries to explore neurophysiological features that related with resilience in IGD patients. Thus, changed EEG activity in the beta or gamma bands indicates more homogeneous features of the IGD pathology, while alpha coherence in the right hemisphere shows us that it is distinct feature associated with resilience in IGD patients. Second, there is possibility that it may get affected by our small cohort of IGD patients. There are significant main and interaction effects in both bands in the present study, further study with large cohort will be needed to clear these inconsistent results.

The present study had several limitations that should be noted. First, only male participants were assessed. Second, the sample size was insufficient to be deemed representative of all IGD patients; because these issues may limit the generalizability of the results, further investigations with larger sample sizes, and including female participants, will be necessary. Third, comorbidities in the IGD patients were not controlled for. However, it has been reported that resilience is strongly related with depression, anxiety, and stress [47]; thus, we suggested moderated mediation models with these factors (depressive symptoms and stress level) which can be regarded as integral to resilience. Despite these limitations, the present study was the first to investigate the association between resilience and the neurophysiological features of IGD patients using a resting-state EEG approach.

The present results demonstrated that IGD patients with low resilience had increased alpha coherence, especially in the right hemisphere. Furthermore, indirect effects of IGD on alpha coherence in the right hemisphere through depressive symptoms and stress level were significant only in IGD patients with the lower degree of resilience. Resilience is a crucial protective trait against the development and maintenance of IGD. Thus, understanding the neurophysiological mechanisms underlying resilience may help to establish effective strategies for treating IGD patients and prevent the occurrence of IGD.

## Figures and Tables

**Figure 1 jcm-08-00049-f001:**
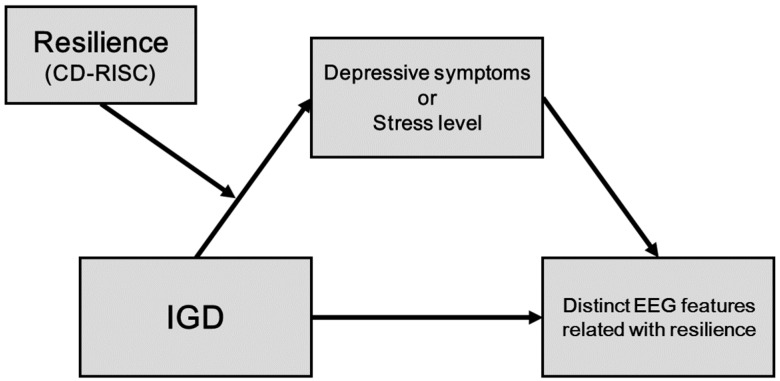
The present study examines whether the indirect effects of Internet gaming disorder (IGD) on resilience-related EEG coherence features through the clinical symptoms (depressive symptoms and stress level) are moderated by resilience.

**Figure 2 jcm-08-00049-f002:**
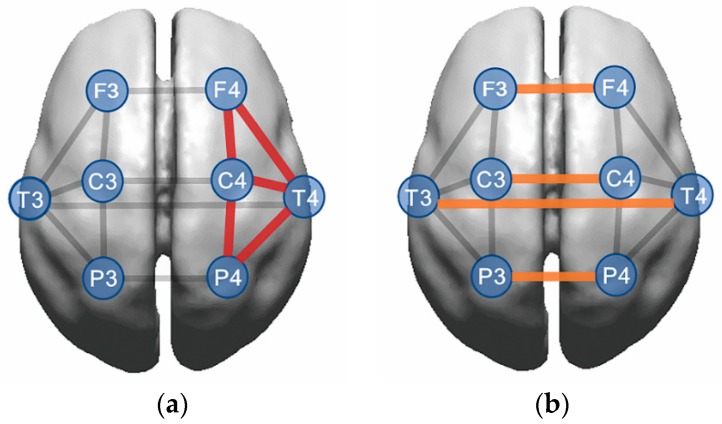
Comparison of coherence among the Internet gaming disorder with low resilience (LowRe-IGD), Internet gaming disorder with high resilience (HighRe-IGD), and health controls (HC) groups. (**a**) Intrahemispheric coherence. The LowRe-IGD group showed greater alpha intrahemispheric coherence, primarily in the right hemisphere, than the HighRe-IGD and HC groups (red line). (**b**) Interhemispheric coherence. The LowRe-IGD group had greater alpha interhemispheric coherence than HC groups (orange line).

**Figure 3 jcm-08-00049-f003:**
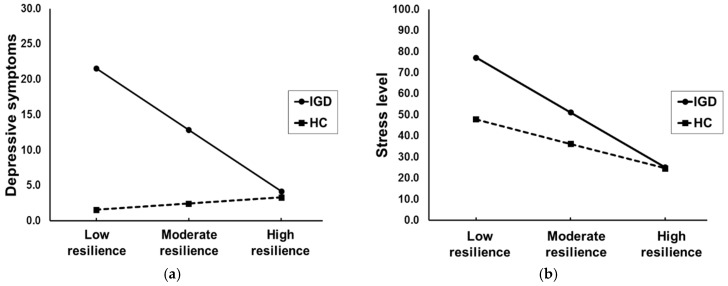
The interaction effects of Internet gaming disorder (IGD) and resilience on (**a**) Depressive symptoms (**b**) Stress level. Greater depressive symptoms and higher stress level than healthy controls (HC) were presented only in IGD patients with low (mean − 1 S.D) and moderate resilience (mean), which it did not correspond to those with high resilience (mean + 1 S.D). Resilience, depressive symptoms, and stress level were assessed by Connor–Davidson resilience scale (CD-RISC), Beck depression inventory-II (BDI), and Psychosocial Well-Being Index (PWI) respectively.

**Table 1 jcm-08-00049-t001:** Demographic and clinical characteristics among LowRe-IGD, HighRe-IGD, and HC groups.

	LowRe-IGD (*n* = 16) Mean ± S.D	HighRe-IGD (*n* = 19) Mean ± S.D	HC (*n* = 36) Mean ± S.D	F	*p*	Post Hoc.
**Demographic data**						
Age	22.13 ± 5.56	25.47 ± 5.04	25.14 ± 3.60	3.037	0.055	
Education (years)	12.44 ± 1.15	13.21 ± 2.02	14.75 ± 1.87	10.833 ***	<0.001	L, H < HC
Game usage in weekday (hour)	8.75 ± 3.99	5.17 ± 3.11	0.88 ± 1.14	50.856 ***	<0.001	HC < H < L
Game usage in weekend (hour)	10.25 ± 3.53	7.55 ± 3.80	1.06 ± 1.01	72.937 ***	<0.001	HC < H < L
**Clinical data**						
IAT	72.63 ± 14.18	57.79 ± 13.13	29.11 ± 7.09	104.552 ***	<0.001	HC < H < L
CD-RISC	26.00 ± 12.81	63.26 ± 9.02	74.14 ± 8.44	137.127 ***	<0.001	L < H < HC
BDI	27.94 ± 9.88	10.68 ± 7.48	3.00 ± 2.99	84.156 ***	<0.001	HC < H < L
PWI	93.25 ± 17.27	47.26 ± 21.67	28.86 ± 12.75	83.903 ***	<0.001	HC < H < L
IQ	102.38 ± 11.40	111.47 ± 15.41	119.75 ± 10.45	11.718 ***	<0.001	L < HC

LowRe-IGD and L = Internet gaming disorder with low resilience; HighRe-IGD and H = Internet gaming disorder with high resilience; HC = healthy controls; S.D = standard deviation; n.s. = not significant; IAT = Young’s Internet addiction test; CD-RISC = Connor–Davidson resilience scale; BDI = Beck Depression inventory-II; PWI = Psychosocial Well-Being Index; IQ = intelligence quotient; *p* < 0.001 ***; The Bonferroni-corrected post-hoc comparison was used (*p* < 0.0167).

**Table 2 jcm-08-00049-t002:** Model effects for coherence among LowRe-IGD, HighRe-IGD, and HC groups.

**Intrahemispheric Coherence**	**Wald** $x^{2}$	**df**	***p***	**Post Hoc**
*Alpha*				
Group	13.515 **	2	0.001	H, HC < L
Group × Region	10.983	10	0.359	
Group × Hemisphere	7.358 *	2	0.025	Right: H, HC < L
Group × Region × Hemisphere	6.590	15	0.968	
*Beta*				
Group	0.908	2	0.635	
Group × Region	26.246 **	10	0.003	n.s
Group × Hemisphere	2.745	2	0.254	
Group × Region × Hemisphere	4.377	15	0.996	
*Gamma*				
Group	1.268	2	0.530	
Group × Region	7.083	10	0.718	
Group × Hemisphere	0.470	2	0.791	
Group × Region × Hemisphere	2.584	15	0.999	
**Interhemispheric Coherence**	**Wald** $x^{2}$	**df**	***p***	**Post Hoc.**
*Alpha*				
Group	9.836 **	2	0.007	HC < L
Group × Region	2.544	6	0.863	
*Beta*				
Group	4.582	2	0.101	
Group × Region	0.953	6	0.987	
*Gamma*				
Group	1.386	2	0.500	
Group × Region	0.990	6	0.986	

LowRe-IGD and L = Internet gaming disorder with low resilience; HighRe-IGD and H = Internet gaming disorder with high resilience; HC = healthy controls; S.D = standard deviation; n.s = not significant; IAT = Young’s Internet addiction test; CD-RISC = Connor–Davidson resilience scale; BDI = Beck Depression inventory-II; PWI = Psychosocial Well-Being Index; IQ = intelligence quotient; *p* < 0.05 *, *p* < 0.01 **; The Bonferroni-corrected post-hoc comparison was used (*p* < 0.0167).

**Table 3 jcm-08-00049-t003:** Moderated mediation model with IGD, alpha coherence in the right hemisphere, depressive symptoms (*n* = 71, bootstrap = 5000).

	**B**	**S.E**	**95% CI**	
			**LLCI**	**ULCI**
**Mediator variable model (Y = depression)**				
IGD (predictor)	10.412 ***	2.441	5.541	15.284
Resilience (Moderator)	0.040	0.128	−0.216	0.297
IGD × Resilience	−0.444 **	0.138	−0.720	−0.169
**Dependent variable model** **(Y = alpha coherence in the right hemisphere)**				
Depression (Mediator)	0.580 **	0.159	0.263	0.897
IGD (predictor)	–2.806	3.664	−10.122	4.509
	**B**	**Boot S.E**	**95% BCa CI**	
			**LLCI**	**ULCI**
**Conditional indirect association on depressive symptoms** **at different values of the resilience (moderator)**				
Low (Mean − 1 S.D)	11.576	3.958	4.507	19.983
Moderate (Mean)	6.039	2.266	2.377	11.375
High (Mean + 1 S.D)	0.502	1.665	−2.072	4.833
Moderated mediation effect	−0.258	0.094	−0.463	−0.093

S.E = standard error; CI = confidence intervals; LLCI = lower limit confidence intervals; ULCI = upper limit confidence intervals; 95% BCa CI = 95% bias-corrected and accelerated bootstrap interval; IGD = Internet gaming disorder; Resilience, depressive symptoms were assessed by Connor–Davidson resilience scale (CD-RISC) and Beck depression inventory-II (BDI); Mean score of CD-RISC was 60.38 and standard deviation (S.D) was 21.48. IQ and years of education were controlled; *p* < 0.01 **, *p* < 0.001 ***.

**Table 4 jcm-08-00049-t004:** Moderated mediation model with IGD, alpha coherence in the right hemisphere, stress level (*n* = 71, bootstrap = 5000).

	**B**	**S.E**	**95% CI**	
			**LLCI**	**ULCI**
**Mediator variable model (Y = stress level)**				
IGD (predictor)	14.891 **	5.280	4.351	25.430
Resilience (Moderator)	−0.540	0.278	−1.094	0.014
IGD × Resilience	−0.669 *	0.299	−1.265	−0.073
**Dependent variable model** **(Y = alpha coherence in the right hemisphere)**				
Stress level (Mediator)	0.179 **	0.059	0.061	0.298
IGD (predictor)	−2.227	3.840	−9.893	5.439
	**B**	**Boot S.E**	**95% BCa CI**	
			**LLCI**	**ULCI**
**Conditional indirect association on stress level at different values of the resilience (moderator)**				
Low (Mean − 1 S.D)	5.244	2.916	1.271	13.060
Moderate (Mean)	2.668	1.557	0.584	6.901
High (Mean + 1 S.D)	0.092	1.252	−2.159	2.962
Moderated mediation effect	−0.120	0.075	−0.326	−0.019

S.E = standard error; CI = confidence intervals; LLCI = lower limit confidence intervals; ULCI = upper limit confidence intervals; 95% BCa CI = 95% bias-corrected and accelerated bootstrap interval; IGD = Internet gaming disorder; Resilience, stress level were assessed by Connor–Davidson resilience scale (CD-RISC) and Psychosocial Well-Being Index (PWI); Mean score of CD-RISC was 60.38 and standard deviation (S.D) was 21.48. IQ and years of education were controlled; *p* < 0.05 *, *p* < 0.01 **.

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
