# Peer review of "Neurophysiological Mechanisms of Resilience as a Protective Factor in Patients with Internet Gaming Disorder: A Resting-State EEG Coherence Study"

_jcm, 2019, doi:10.3390/jcm8010049_

Round 1

Reviewer 1 Report

The authors have presented the findings from a study to examine resting-state EEG in patients with internet gaming disorder compared to healthy controls. They further explore the role of resilience in IGD. The study has been carefully designed and conducted, and the results are interesting and clearly presented. However, I wonder how the results will guide treatment strategies for patients with IGD. I also have some minor concerns about the population used and the sample size. 

Comments: 

The Introduction is generally well written, but there are some grammatical issues on lines 52 through 58 that need to be addressed. The findings reported in this section are unclear. The Introduction should also provide more information (1-2 sentences) about how resilience has been measured in these previous studies. 

Were healthy controls matched to patients with IGD for age or any other demographic characteristic? What is the justification for including only males in the study? 

How was the sample size determined, and is it sufficient for all the reported subgroup and mediation analyses?

Lines 240-242 of the Results seem to contain instructions that are not part of the main paper and should be removed. 

The authors indicate that years of education and IQ score were adjusted for in the model (lines 213-214) because they differed among groups. According to Table 1, all clinical data also differed by group. Was this also adjusted for in the model?  

The results are interesting, but I wish the authors had spent more time discussing the implications of these results in terms of treatment. How does knowing that low resilient IGD patients have higher alpha coherence guide treatment options? What treatment options are currently available?

It is difficult to distinguish the symbols in Figure 3. Can the symbols be made bigger? 

Author Response

#. Reviewer 1

Comments and Suggestions for Authors:

The authors have presented the findings from a study to examine resting-state EEG in patients with internet gaming disorder compared to healthy controls. They further explore the role of resilience in IGD. The study has been carefully designed and conducted, and the results are interesting and clearly presented. However, I wonder how the results will guide treatment strategies for patients with IGD. I also have some minor concerns about the population used and the sample size. 

Comments: 

1.  The Introduction is generally well written, but there are some grammatical issues on lines 52 through 58 that need to be addressed. The findings reported in this section are unclear. The Introduction should also provide more information (1-2 sentences) about how resilience has been measured in these previous studies. 

>> Response 1: I appreciate your critical comments. As you pointed out, we realized that our previous version of manuscript had some weaknesses with giving explanation of resilience as a protective factor for IGD. We revised line 52 to 58 grammatically for readers to understand clearly. Additionally, we added several studies that revealed resilience as a protective factor of IGD to address the aim of present study. Revised paragraph is as below:

Recently, researchers and clinicians have shifted their focus from risk factors for IGD to protective factors for the disorder, with the aim of preventing its occurrence and aiding the recovery of IGD patients. A review of longitudinal studies on adolescents and young adults demonstrated that higher levels of self-control and self-esteem, in addition to satisfaction of basic psychological needs, protect against problematic Internet use [6]. Choi et al. [7] reported that character strength of courage indicating encompasses vitality and bravery has a negative relationship with IA.

Meanwhile, resilience is receiving attention because it regarded as one of the prominent protective factors for IGD. Resilience can be defined as the personal qualities that enable one to thrive in the face of adversity throughout life [8]. Resilience involves rapid recovery from mental health disturbances following exposure to adversity [9]. Resilience plays an important role in life satisfaction [10]. Resilience is an important protective factor for the pathophysiology of addictive disorders [11] and is a strong protective factor against IGD. A previous study showed that higher the level of the participants' resilience assessed by Connor–Davidson Resilience Scale (CD-RISC), the lower the level of their internet addiction [12,13]. In a multiple mediation study with children showed that resilience was strongly associated with Internet addiction directly, and also indirectly, through peer relationship and depression. This study suggests that resilience is as a predictor of IA [14].

2. Were healthy controls matched to patients with IGD for age or any other demographic characteristic? What is the justification for including only males in the study? 

>> Response 2: We only matched sex with patients with IGD. Healthy controls in the present study were recruited from the local community and universities. Experienced clinical psychiatrist screened their history of any psychiatric disorder and internet usage.

The reason that we assessed only male participants is that it was known that IGD is prevalent in male compared with female in South Korea. A survey on Internet addiction in 2014 showed that male accounted for 58.8% of the composition of high-risk for Internet addiction (National Information Society Agency, 2015). Therefore, we preferentially focused on studying with male patients. However, as we mentioned in the limitation, further investigations with female participants, will be necessary for generalizing the present results.

3. How was the sample size determined, and is it sufficient for all the reported subgroup and mediation analyses?

>> Response 3: Sample size was determined by previous study (Park, 2017*; Park, 2018**). Since we divided IGD patients into subgroups, we tried to compensate small sample in each subgroup by using generalized estimating equation (GEE). GEE is the estimates of the regression parameters are asymptotically unbiased even if the correlation structure is misspecified, although their small sample properties are not known (Paul, 2014***).

* Park, S.M.; Lee, J.Y.; Kim, Y.J.; Lee, J.-Y.; Jung, H.Y.; Sohn, B.K.; Choi, J.-S. Neural connectivity in Internet gaming disorder and alcohol use disorder: a resting-state EEG coherence study. Sci. Rep. 2017, 7, 1333.

** Park, S.; Ryu, H.; Lee, J.-Y.; Choi, A.; Kim, D.-J.; Kim, S.N.; Choi, J.-S. Longitudinal Changes in Neural Connectivity in Patients With Internet Gaming Disorder: A Resting-State EEG Coherence Study. Frontiers in Psychiatry 2018, 9, doi:10.3389/fpsyt.2018.00252.

*** Nam et al. The Role of Resilience in Internet Addiction among Adolescents between Sexes: A Moderated Mediation Model. J Clin Med, 2018, 7.8: 222.

**** Paul, S.; Zhang, X. Small sample GEE estimation of regression parameters for longitudinal data. Statistics in medicine, 2014, 33, 3869-3881.

4. Lines 240-242 of the Results seem to contain instructions that are not part of the main paper and should be removed. 

>> Response 4: Thank you for your helpful comments. Line 240 to 242 of Results were removed. 

5. The authors indicate that years of education and IQ score were adjusted for in the model (lines 213-214) because they differed among groups. According to Table 1, all clinical data also differed by group. Was this also adjusted for in the model?  

>> Response 5: As you mentioned, clinical variables including Young’s Internet addiction test (YIAT), Connor-Davidson resilience scale (CD-RISC), Beck Depression inventory-Ⅱ (BDI), Psychosocial Well-Being Index (PWI), and intelligence quotient (IQ) were differed among groups. In those variables, CD-RISC, BDI, and PWI were inserted as independent variable and mediated variables respectively. Age and years of education were controlled in the models as well. However, YIAT was not adjusted in the model since it is the important feature that distinguish between IGD patients and healthy controls. Game usage was assessed for the severity of IGD symptoms.

6. The results are interesting, but I wish the authors had spent more time discussing the implications of these results in terms of treatment. How does knowing that low resilient IGD patients have higher alpha coherence guide treatment options? What treatment options are currently available?

>> Response 6: Thank you for your critical comments. The present results can be applied to neuromodulation treatments including repetitive transcranial magnetic stimulation (rTMS) or transcranial Direct-Current Stimulation (tDCS). Many studies have applied those neuromodulation methods to reveal therapeutic effects on addictive disorders (Bari, 2018*; Azevedeo, 2018**). Recently, it is known that considering EEG coherence with those treatments would improve therapeutic effects. A rTMS study with patients with resistant depression showed that resting theta coherence was higher in responders to rTMS for depression and it showed potential for predictive response to rTMS treatment for depression (Bailey, 2019***). In the study with patients with disorders of consciousness, therapeutic effects of tDCS rely on modulation of fronto-parietal coherence in patients with residual consciousness (Cavinato, 2018****). Based on previous results, it is expected that applying neuromodulation treatments considering EEG coherence may improve therapeutic effects on IGD patients with low resilience. However, we cannot contain those treatment options in the present manuscripts because there are insufficient findings to apply those methods to IGD patients. Therefore, we are planning to further investigations that studies how neuromodulations effect on IGD patients and how EEG coherence is differentiated by resilience. 

*Bari, A.; De Cisare, J.; Babayan, D.; Runcie, M.; Sparks, H.; Wilson, B. Neuromodulation for substance addiction in human subjects: a review. Neurosci Biobehav Rev, 2018.

** Azevedo, C. A.; Mammis, A. (2018). Neuromodulation therapies for alcohol addiction: a literature review. Neuromodulation: Technology at the Neural Interface, 2018, 21, 144-148.

*** Bailey N.; Hoy K.; Rogasch N.; Thomson R; McQueen S; Elliot D; Sullivan C; Fulcher B; Daskalakis Z; Fitzgerald P. Differentiating responders and non-responders to rTMS treatment for depression after one week using resting EEG connectivity measures. J Affect Disord, 2019, 1;242:68-79.

**** Cavinato, M.; Genna, C.; Formaggio, E.; Gregorio, C.; Storti, S. F.; Manganotti, P.; Piccione, F. (2018). Behavioural and electrophysiological effects of tDCS to prefrontal cortex in patients with disorders of consciousness. Clin Neurophysiol, 2018.

7. It is difficult to distinguish the symbols in Figure 3. Can the symbols be made bigger? 

>> Response 7: We would like to thank you for your comment. Figure 3 was revised with bigger symbols.

Reviewer 2 Report

In the present study the Authors aimed to determine the neurophysiological correlates of resilience, via resting-state EEG coherence analysis, in IGD patients and HC. Overall, I found the paper very interesting, well written and scientifically sound. I have only few comments:

1) In the Introduction, I would suggest to briefly mention that IGD may be often related to dissociative experiences and alexithymia. See the studies of De Berardis et al that should be cited.

2) How intellectual disability was assessed?

3) I suggest to add a Table with DSM-V with IGD criteria 

Author Response

#. Reviewer 2

Comments and Suggestions for Authors:

In the present study the Authors aimed to determine the neurophysiological correlates of resilience, via resting-state EEG coherence analysis, in IGD patients and HC. Overall, I found the paper very interesting, well written and scientifically sound. I have only few comments:

1. In the Introduction, I would suggest to briefly mention that IGD may be often related to dissociative experiences and alexithymia. See the studies of De Berardis et al that should be cited.

>> Response 1: We appreciate your helpful comments. We mentioned the association between alexithymia and IGD in the revised manuscripts line 51-54 as below:

Especially, patients with alexithymia reported the high likelihood of IA. They had difficulty in identifying feelings, higher dissociative experiences, lower self-esteem, and higher impulse dysregulation were associated with higher emergence of IA [6].

2. How intellectual disability was assessed?

>> Response 2: Thank you for your helpful comment. Experienced clinical psychiatrist screened had a history of significant head injury, seizure, intellectual disability, or psychotic or neurological disorders. Intellectual disability was assessed by the Korean version of the Wechsler Adult Intelligence Scale (WAIS-IV) to estimate IQ levels. This sentence was described in the revised manuscript (line 175) as bellows:

The Korean version of the Wechsler Adult Intelligence Scale [37] was administered to all participants to estimate IQ levels.

3. I suggest to add a Table with DSM-V with IGD criteria 

>> Response 3: As your suggestion, we added a table of IGD criteria in the Supplementary Tables.
